# Monitoring cytosolic $H_2O_2$ fluctuations arising from altered plasma membrane gradients or from mitochondrial activity

Mercè Carmona [1,3], Laura de Cubas[1,3], Eric Bautista[1], Marta Moral-Blanch[1], Iria Medraño-Fernández[2], Roberto Sitia [2], Susanna Boronat[1], José Ayté [1] & Elena Hidalgo [1]*

Genetically encoded probes monitoring $H_2O_2$ fluctuations in living organisms are key to decipher redox signaling events. Here we use a new probe, roGFP2-Tpx1.C169S, to monitor pre-toxic fluctuations of peroxides in fission yeast, where the concentrations linked to signaling or to toxicity have been established. This probe is able to detect nanomolar fluctuations of intracellular $H_2O_2$ caused by extracellular peroxides; expression of human aquaporin 8 channels $H_2O_2$ entry into fission yeast decreasing membrane gradients. The probe also detects $H_2O_2$ bursts from mitochondria after addition of electron transport chain inhibitors, the extent of probe oxidation being proportional to the mitochondrial activity. The oxidation of this probe is an indicator of steady-state levels of $H_2O_2$ in different genetic backgrounds. Metabolic reprogramming during growth in low-glucose media causes probe reduction due to the activation of antioxidant cascades. We demonstrate how peroxiredoxin-based probes can be used to monitor physiological $H_2O_2$ fluctuations.

---

[1] Oxidative Stress and Cell Cycle Group, Universitat Pompeu Fabra, C/Dr. Aiguader 88, 08003 Barcelona, Spain. [2] Protein Transport and Secretion Unit, Division of Genetics and Cell Biology, Istituto di Ricovero e Cura a Carattere Scientifico (IRCCS) Ospedale San Raffaele, Università Vita-Salute San Raffaele, 20132 Milan, Italy. [3] These authors contributed equally: Mercè Carmona, Laura de Cubas. *email: elena.hidalgo@upf.edu

Aerobic organisms inevitably produce reactive oxygen species as a side effect of sequential one-electron reduction of oxygen during respiration, among other processes. One of these species, hydrogen peroxide ($H_2O_2$), has been traditionally linked to the toxicity associated to aerobic metabolism. However, $H_2O_2$ also clearly participates in signaling events, by both regulating different physiological processes and also activating antioxidant cascades (for reviews, see refs. [1–3]).

The signaling role of $H_2O_2$ is often associated to the direct oxidation of cysteine (Cys) residues via reversible modifications, normally disulfide bridges. Reversibly oxidized thiols are later on reduced by the thioredoxin (Trx) and glutaredoxin (Grx) systems and reduced cofactor[4,5]. Very few Cys in proteins fulfill the prerequisites for fast and efficient oxidation by moderate fluctuations of peroxides. These special Cys residues are known as thiol switches, and belong to 'true' $H_2O_2$ protein sensors[6]. Among the proteins unambiguously responding to pre-toxic doses of $H_2O_2$ to induce antioxidant responses are the prokaryotic transcription factor OxyR[7] and the *Schizosaccharomyces pombe* Tpx1[8,9], which belongs to the family of $H_2O_2$ scavengers known as peroxiredoxins (Prxs). OxyR and Prxs display high reactivity to peroxides, with reaction rates in the range of $10^5$–$10^7\,M^{-1}\,s^{-1}$[10]. The gain-of-function of the $H_2O_2$-activated sensor, acting either as a transcription factor (in the case of OxyR) or as a redox transducer (in the case of Tpx1), has to be sustained to be capable of inducing an adaptive response. Thus, we have recently proposed that $H_2O_2$-oxidized thiols in real protein sensors such as OxyR should also fulfill the requisite of displaying slow reduction rates. On the contrary, in the case of the moonlighting $H_2O_2$ peroxidase Tpx1 its reduction rates are very fast; only when the Trx system becomes transiently exhausted Tpx1 switches from a $H_2O_2$ scavenger to a signal transducer toward the transcription factor Pap1[11].

The intracellular levels of $H_2O_2$ are tightly controlled by scavenging activities, so that fluctuations over a certain threshold can drive adaptation responses. Measuring those variations has been the focus of attention of many redox biology laboratories for decades. While most measurements are still based on the use of permeable fluorescent dyes which respond with different specificity and sensitivity to $H_2O_2$, their use is controversial[12]. Over a decade ago[13], several groups decided to investigate the use of protein-based reporters to measure intracellular redox potentials and, later on, $H_2O_2$. Thus, two major families of genetically encoded redox reporters have been developed: the reduction-oxidation-sensitive green fluorescent protein (roGFP)-based proteins[13] and HyPer and derivatives[14] (for excellent reviews on the use of these genetic reporters, see refs. [15,16]).

HyPer was engineered to sense and report intracellular fluctuations of $H_2O_2$. It is based on the transcription factor OxyR, which is activated by $H_2O_2$ to trigger an antioxidant response. The two Cys residues of OxyR involved in $H_2O_2$-dependent disulfide bond formation are located in positions 199 and 208[7]. In HyPer, the circularly permuted yellow fluorescent protein (cpYFP) is inserted between residues 205 and 206 of OxyR, so that disulfide bond formation generates a conformational change in cpYFP affecting its fluorescent properties. Thus, reduced HyPer has two excitation peaks, with maximal values at 420 and 500 nm, and one emission peak at 516 nm. Upon oxidation with peroxides, the magnitude of the 420 nm peak decreases concomitant with a proportional increase in the magnitude of the 500 nm peak[14]. These ratiometric changes are specific for $H_2O_2$, and other oxidants such as superoxide, oxidized glutathione or other reactive oxygen species are not able to induce them in vitro[14].

roGFP has several substitutions on surface-exposed domains, so that disulfide bonds can be formed in response to oxidants and

change the fluorescence properties of the protein in an analogous manner to HyPer[13]. The specificity and sensitivity of roGFP to sense oxidants has been improved during the last years by fusing real $H_2O_2$ protein sensors to this fluorescent reporter. Thus, a variant of roGFP has been fused to the *Saccharomyces cerevisiae* glutathione peroxidase Orp1[17] or to a Prx lacking its resolving Cys[18], yielding the chimeric proteins roGFP2-Orp1 and roGFP2-Tsa2$\Delta C_R$. Both, but specially roGFP2-Tsa2$\Delta C_R$, have been shown to monitor intracellular $H_2O_2$ levels with unprecedented sensitivity in vivo[18,19].

Often, characterization of $H_2O_2$ fluorescent protein sensors is performed after application of extracellular peroxides, but the exact correlation between extracellular and intracellular peroxide concentrations is not known in most biological systems. In fission yeast, we have recently calculated that the gradient of extracellular-to-intracellular peroxides through fission yeast membranes is around 40:1, but intracellular $H_2O_2$ scavenging activities such as Tpx1 can enhance this gradient up to 300:1[11]. In fact, the responsibility to maintain non-toxic peroxide levels relies exclusively on this protein: cells lacking Tpx1 display strong growth defects in the presence of oxygen[20], due to low micromolar steady-state levels of $H_2O_2$ (specifically, 0.3 μM)[11]. This suggests that Tpx1 has high specificity and sensitivity for $H_2O_2$. With this knowledge in mind, we have expressed in the cytosol of fission yeast a new and ultrasensitive genetically encoded reporter based on this exquisitely sensitive Prx, Tpx1. Similar to its budding yeast ortholog roGFP2-Tsa2$\Delta C_R$, the probe is able to sense moderate fluctuations of $H_2O_2$ induced by genetic or environmental interventions in fission yeast cells. Its oxidation by extracellular peroxides can be exacerbated by overexpression of human aquaporin 8, which facilitates $H_2O_2$ channeling across membranes. The basal level of oxidation of roGFP2-Tpx1.C169S in wild-type cells is around half of its maximum levels, while cells lacking Tpx1 display 70–80% oxidation of the probe, indicating that this extent of oxidation is due to the toxic low micromolar levels. Furthermore, the cytosolic probe is able to sense the leakage of $H_2O_2$ from the mitochondria upon addition of the electron transport chain inhibitor antimycin A. Using this Prx-based sensor, we demonstrate that it facilitates quantitative measurements of dynamic and pre-toxic $H_2O_2$ fluxes arising from the extracellular milieu or from the mitochondria.

## Results

**Expression of HyPer and roGFP derivatives in *S. pombe*.** Many genetically encoded reporters have been designed and analyzed in different model systems, which makes it difficult to interpret and compare the results. We decided to create our own probes, based on fission yeast's Gpx1 and Tpx1 peroxidases, with two main goals: first, to unambiguously compare the sensitivity of previous and new sensors in the same model system; and second, to correlate specific intracellular peroxide concentrations, previously established in fission yeast[11], with the oxidation of different cytosolic reporters.

We first expressed HyPer[14], roGFP2[13], and Grx1-roGFP2[21] in the cytosol of *S. pombe* cells under the control of the constitutive promoter *sty1*[22], using episomal plasmids. The high concentration of the reporters allowed us to monitor probe oxidation directly in exponentially growing cultures, without centrifugation or media transfer. We applied extracellular concentrations of peroxides ranging from 1 to 25 μM (pre-toxic), 50 to 100 μM (signaling toward the antioxidant Pap1 cascade), and 0.2 to 1 mM (toxic, halts growth of wild-type cells), corresponding to intracellular concentrations of 3–80 nM, 0.2–0.3 μM, and 0.7–3 μM, respectively[11]. As shown in Fig. 1a–c and Supplementary Fig. 1, the three probes HyPer, roGFP2, and Grx1-roGFP2

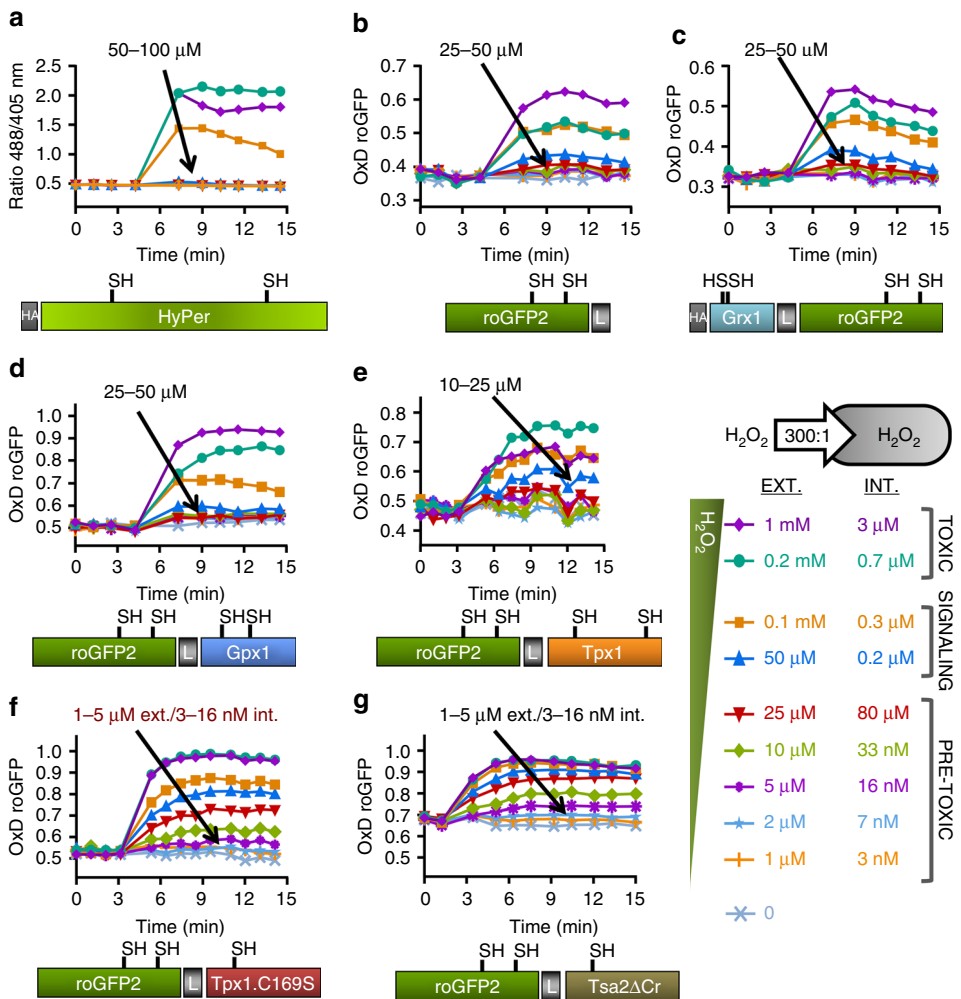

**Fig. 1** Oxidation of HyPer and roGFP derivatives upon extracellular $H_2O_2$. Wild-type strain HM123 was transformed with plasmids p605, p406, p379, p429, p407, p407.C169S, and p676, to express HyPer (**a**), roGFP2 (**b**), Grx1-roGFP2 (**c**), roGFP2-Gpx1 (**d**), roGFP2-Tpx1 (**e**), roGFP2-Tpx1.C169S (**f**), and roGFP2- Tsa2$\Delta C_R$ (**g**), respectively. The indicated extracellular (EXT.) concentrations of $H_2O_2$, classified into 'Toxic' (able to temporarily halt growth), 'Signaling' (capable to trigger the antioxidant Pap1 cascade) or 'Pre-toxic' (unable to activate Pap1), were directly added to MM cultures at an $OD_{600}$ of 1 in 96-well imaging plates, and growth proceeded at 30 °C with shaking. The corresponding intracellular peroxide concentrations, after applying a 300:1 membrane gradient, are also indicated (INT.). The degree of probe oxidation (amount of probe oxidized per 1) is indicated in the Y axis (OxD roGFP), except for HyPer (**a**), for which the ratio 488/405 nm is shown. For each panel, data from three biological replicates are shown, with error bars (S.D.) displayed in Supplementary Fig. 1

are oxidized in response to $H_2O_2$ treatments; the minimum concentrations of extracellular peroxides triggering probe oxidation are 50, 25, and 25 μM, respectively. HyPer displays a maximum oxidation with extracellular 0.2 mM (Fig. 1a).

We then fused *S. pombe* Gpx1 and Tpx1 to roGFP2, and determined that the $H_2O_2$ thresholds capable of inducing probe oxidation were 25 and 10 μM, respectively (Fig. 1d, e), barely improving the values obtained with the previously published roGFP reporters.

**roGFP2-Tpx1.C169S is very sensitive to $H_2O_2$ addition**. As indicated in the Introduction, we have recently established that Tpx1 is not only exquisitely sensitive to oxidation by peroxides, but also very efficiently recycled by the Trx system[11]. Thus, we fused Tpx1.C169S to roGFP2. This mutant protein, which lacks the resolving Cys, is capable of interacting with $H_2O_2$ but the peroxidatic Cys48-SOH cannot react with the resolving Cys to form a disulfide. Therefore, this mutation eliminates Trx

competition and may enhance the oxidation transfer from Cys48-SOH to roGFP2. As shown in Fig. 1f, roGFP2-Tpx1.C169S is able to sense extracellular concentrations as low as 1–5 μM $H_2O_2$; this sensitivity is similar to that of the previously characterized reporter roGFP2-Tsa2$\Delta C_R$, derived from the *S. cerevisiae* peroxiredoxin Tsa2, expressed in fission yeast (Fig. 1g). If we consider a gradient of extracellular-to-intracellular peroxides of 300-to-1[11], an intracellular concentration of as little as 3–16 nM $H_2O_2$ can be monitored with both Prx-based probes. On the contrary, HyPer, roGFP, or Grx1-roGFP can only detect concentrations of $H_2O_2$ high enough to activate antioxidant cascades, or even halting the growth of cell cultures. As previously described in vitro for roGFP2-Tsa2$\Delta$CR, both Prx-based chimeras are also very sensitive in vivo to other oxidants such as tert-butyl hydroperoxide, menadione (generates in vivo superoxide, which is dismuted to $H_2O_2$) and sodium hypochlorite (Supplementary Fig. 2a), but barely or do not respond to high concentrations of a nitric oxide donor, oxidized GSH, or dehydroascorbic acid (Supplementary Fig. 2b). The protein sequence of roGFP2-Tpx1.C169S is

**Effect of AQP8 expression on roGFP2-Tpx1.C169S oxidation**. We have demonstrated that the diffusion of $H_2O_2$ across fission yeast membranes is limited, and that scavenging mainly by Tpx1 enhances the gradients up to 300:1[11]. We tested whether expression of human AQP8 in *S. pombe* could decrease those gradients, by monitoring oxidation of cytosolic roGFP2-Tpx1. C169S. Aquaporins (AQPs) are diffusion facilitators for non-charged and partially polar molecules such as water or glycerol. Recent studies have demonstrated that some mammalian AQPs, such as AQP8[23], also mediate the transport of $H_2O_2$ across human cell membranes. As shown in Fig. 2a and Supplementary Fig. 3a, expression of AQP8 in fission yeast facilitates $H_2O_2$ entry by a factor of at least five times, and probe oxidation can be detected with as little as 0.2–1 μM extracellular peroxides. Con-comitantly, expression of AQP8 in wild-type cells (Fig. 2b) or in strain Δ*pap1*, lacking an antioxidant transcription factor[8] (Supplementary Fig. 3b), enhances sensitivity to peroxides on solid plates, without affecting the tolerance to other stressors such as potassium chloride. Importantly, expression of AQP4, which does not channel $H_2O_2$ across membranes, does not affect the toler-ance of fission yeast to peroxides, and probe oxidation does not differ from that of wild-type cells (Supplementary Fig. 3a, b).

**roGFP2-Tpx1.C169S scavenges $H_2O_2$ and affects cell physiol-ogy**. We have previously shown that cells lacking Tpx1 or expressing the catalytically inactive Tpx1.C48S mutant display severe phenotypes to grow in the presence of oxygen[11,20]. Nevertheless, Tpx1.C169S can sustain aerobic growth, probably with GSH supplying a thiol group to substitute for Cys169[20]. To test whether expression of the probe roGFP2-Tpx1.C169S affects cellular $H_2O_2$ homeostasis, we transformed strain Δ*tpx1* with episomal and integrative plasmids containing the chimeric gene *roGFP2-tpx1.C169S*. As shown in Supplementary Fig. 4, high levels of roGFP2-Tpx1.C169S are able to largely suppress the aerobic growth defects of cells lacking Tpx1. On the contrary, the lower levels accomplished from the integrative plasmid only partially restore aerobic growth.

**roGFP2-Tpx1.C169S $OxD_0$, an indicator of $H_2O_2$ steady-state levels**. Cells lacking Tpx1 display aerobic growth defects due to the absence of the main scavenger of peroxides during aerobic growth[20]. We aimed at studying whether the new probe would be able to sense the enhanced steady-state levels of peroxides in this strain background. We compared the basal and extracellular $H_2O_2$-triggered oxidation ratios of both families of probes, HyPer and roGFP2-Tpx1.C169S. As shown in Fig. 3a and Supplemen-tary Fig. 5, the basal oxidation levels of HyPer expressed in cells lacking Tpx1 were identical to those observed in wild-type cells. Interestingly, HyPer expressed in strain Δ*tpx1* was now capable of sensing as little as 10–25 μM extracellular $H_2O_2$. This reflects the fact that scavenging by Tpx1 is now jeopardized and only a 40-to-1 gradient of extracellular-to-intracellular peroxides applies in this strain background due only to permeability. In wild-type cells, both permeability and $H_2O_2$ scavenging generate a 300-to-1 step gradient[11].

roGFP2-Tpx1.C169S expressed in Δ*tpx1* is now able to sense concentrations of peroxides below 1 μM, as shown in Fig. 3b (right panel). But, most importantly, the basal level of oxidation of the probe, $OxD_0$, is also significantly altered, with only 50–60% of the probe being oxidized in a wild-type background (Fig. 3b,

left panel) and moving up to 70–80% in strain Δ*tpx1* (Fig. 3b, right panel). Importantly, the enhanced $OxD_0$ of this probe in Δ*tpx1* is due to elevated $H_2O_2$, since Grx1-roGFP2, designed to sense redox changes in GSH, does not exhibit this increase (Supplementary Fig. 5c). We conclude that basal oxidation levels, $OxD_0$, of roGFP2-Tpx1.C169S can be used to monitor $H_2O_2$ steady-state levels in different strain backgrounds.

**Effect of genetic mutations on roGFP2-Tpx1.C169S redox state**. We have demonstrated that steady-state $H_2O_2$ levels, enhanced in Δ*tpx1*, can change basal oxidation of the roGFP2-Tpx1.C169S probe, $OxD_0$ (Fig. 3b). The steady-state oxidation of the probe, however, may also depend on the thiol reducing capacity of the cell. We deleted individual genes coding for components of the Trx and Grx thiol reducing systems. As reported before for other roGFP derivatives[18], the Grx system is the major reductant of the roGFP disulfide bond: the basal steady-state levels of oxidized roGFP2-Tpx1.C169S are sig-nificantly enhanced in cells lacking Pgr1 (GSH reductase) or Grx1, the main cytosolic glutaredoxin (Fig. 4a and Supplementary Fig. 6a).

Regarding genetic mutations in the cytosolic Trxs, Trx1, and Trx3, they are not expected to exert a major impact on roGFP or on Tpx1.C169S redox state, since the resolving Cys of the peroxidase is missing (see Introduction)[20]. Nevertheless, these Trxs are required to recycle Tpx1[24], and therefore extracellularly added $H_2O_2$ may not be efficiently scavenged in cells lacking both Trxs. As shown in Fig. 4b, the roGFP2-Tpx1.C169S reporter is not significantly affected under basal conditions in strain Δ*trx1*. Δ*trx1* Δ*trx3* cells display an $OxD_0$ of 0.8, confirming that the absence of both Trxs have a larger impact on Tpx1 recycling[24]. From these experiments, we conclude that the basal oxidation level of roGFP2-Tpx1.C169S can be affected by alterations of the GSH/Grx1 system at the level of probe reduction, and that it responds directly to $H_2O_2$ perturbations in the case of Trx mutants (Fig. 4c).

We next monitored whether probe oxidation upon extra-cellular $H_2O_2$ was reversible. We applied different concentrations of $H_2O_2$ to wild-type and Δ*ctt1* strains, lacking catalase, and monitored probe oxidation during 7 h (Fig. 4d). In a wild-type background, 2–2.5 h are sufficient for full probe reduction after applying extracellular concentrations of peroxides lower than 100 μM (Fig. 4d, upper panel). These kinetics are very similar in cells lacking Ctt1 (Fig. 4d, lower panel). The oxidized probe in wild-type cells requires longer times (3 and 6 h, respectively) to return to basal OxD values after 0.2 and 1 mM $H_2O_2$ stress, which fits with the effect of those toxic concentrations halting culture growth during 1–5 h[11,22]. Importantly, cell cultures of strain Δ*ctt1* cannot resume growth after 0.2 and 1 mM extracellular stresses, and the roGFP2-Tpx1.C169S probe remains partially (0.2 mM) or fully oxidized (1 mM $H_2O_2$) more than 7 h after stress imposition (Fig. 4d, lower panel). In conclusion, the roGFP2-Tpx1.C169S probe is a reversible indicator of intracellular concentrations of peroxides.

**roGFP2-Tpx1.C169S can detect mitochondrial $H_2O_2$ produc-tion**. An important source of $H_2O_2$ production is the mito-chondrial electron transport chain (ETC). We tested whether cytosolic roGFP2-Tpx1.C169S is able to sense $H_2O_2$ produced at and diffused out of the mitochondria. We treated cells with antimycin A (ANT), an inhibitor of complex III, which exacer-bates accidental reactive oxygen species (ROS) production in the mitochondria due to the accumulation of reduced upstream ETC components[25]. We first monitored cytosolic HyPer oxidation upon ANT addition in both a wild-type strain and in cells lacking

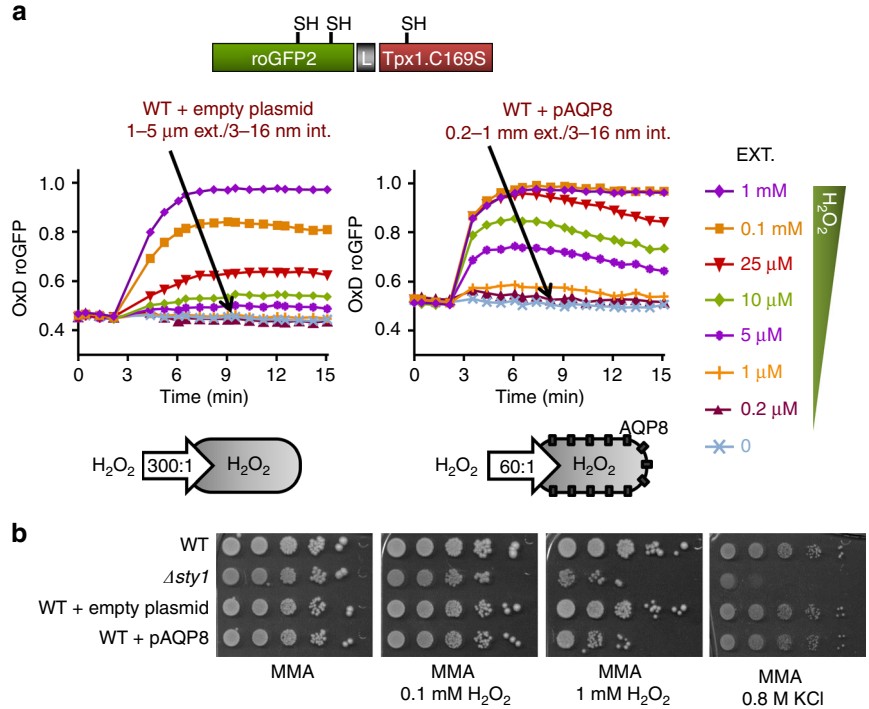

**Fig. 2** Oxidation of roGFP2-Tpx1.C169S by $H_2O_2$ is exacerbated by expression of human AQP8. **a** Wild-type strain PN513 was co-transformed with plasmid p407.C169S, coding for roGFP2-Tpx1.C169S, and an empty plasmid (left) or plasmid p675 to trigger the constitutive expression of AQP8. $H_2O_2$ treatments were performed and oxidation of the reporter estimated as described in Fig. 1. Data from three biological replicates are shown, with error bars (S.D.) displayed in Supplementary Fig. 3. **b** Expression of AQP8 in fission yeast decreases wild-type tolerance to peroxides. Serial dilutions of MM cultures of strains 972 (WT), AV18 (Δsty1), and 364 carrying pREP.42× (WT + empty plasmid) or p675 (WT + pAQP8) were spotted on MM agar plates and the indicated concentrations of $H_2O_2$ or potassium chloride (KCl), and grown for 3–4 days at 30 °C

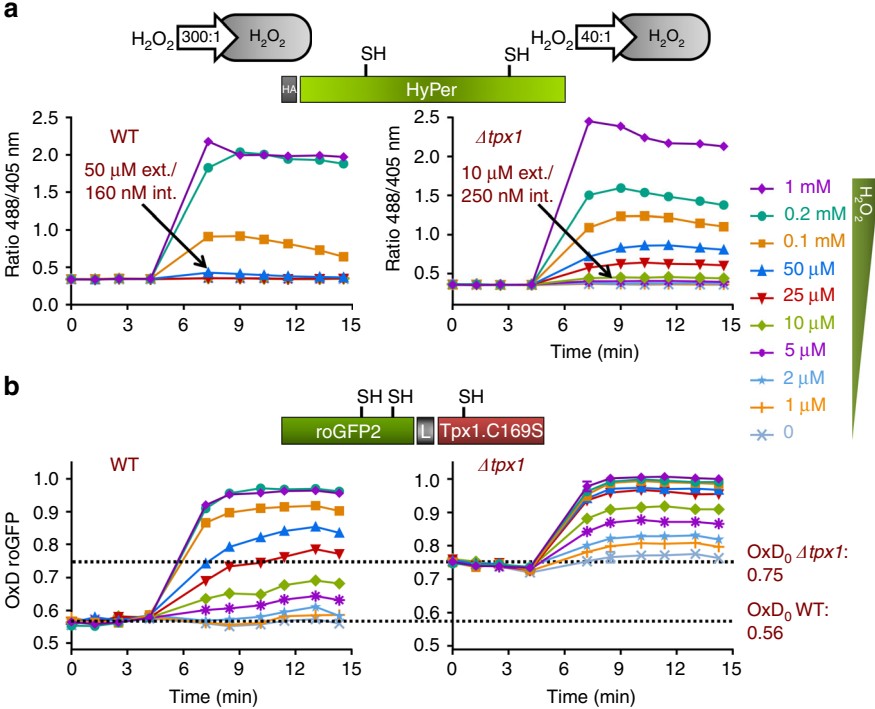

**Fig. 3** Basal levels of oxidation, $OxD_0$, of roGFP2-Tpx1.C169S are elevated in cells lacking Tpx1. Strains HM123 (WT) and SG5 (Δtpx1) were transformed with plasmids p605, coding for HyPer (**a**) and p407.C169S, coding for roGFP2-Tpx1.C169S (**b**) and $H_2O_2$ treatments were performed and oxidation of the reporters estimated as described in Fig. 1. For each panel, data from three biological replicates are shown, with error bars (S.D.) displayed in Supplementary Fig. 5

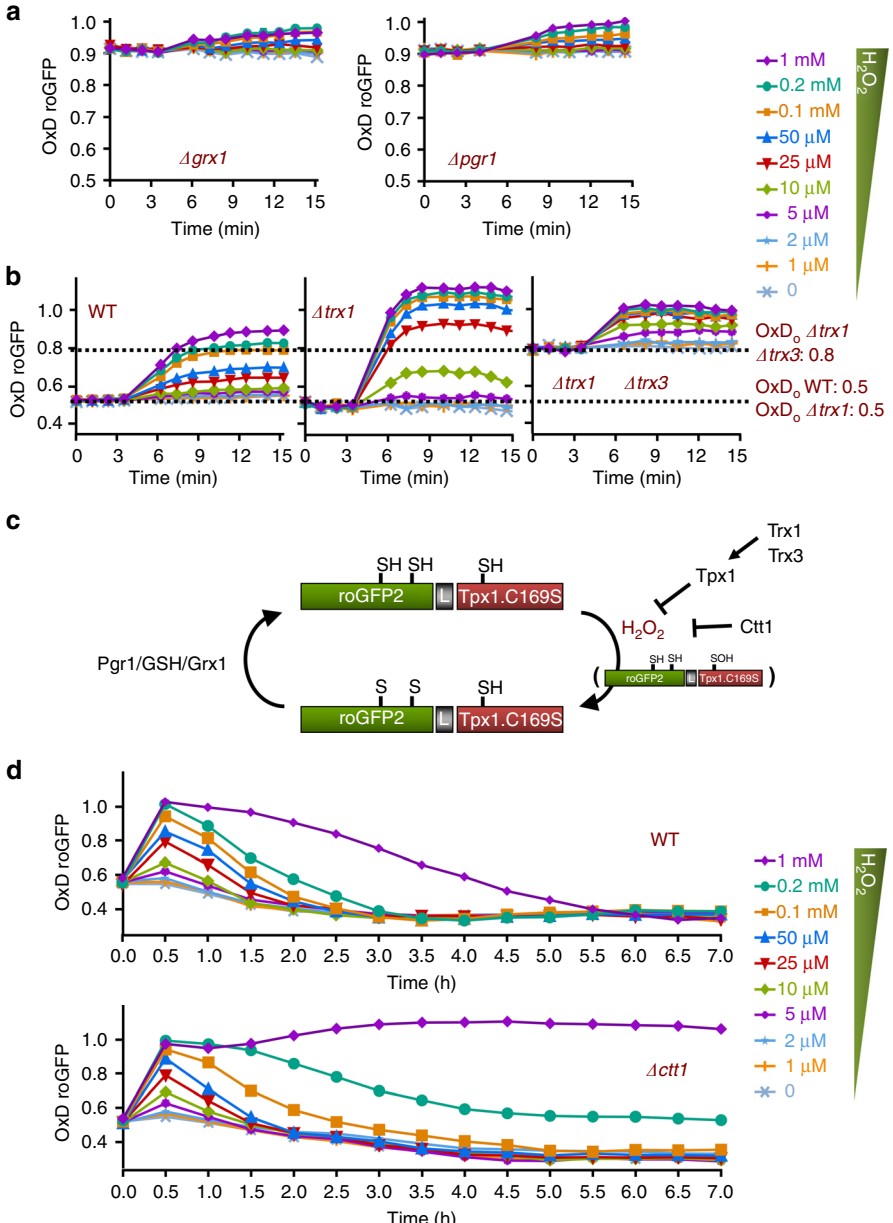

**Fig. 4** Effect of genetic mutations on roGFP2-Tpx1.C169S oxidation. Strains SB36 (Δ*grx1*) and AD88 (Δ*pgr1*) (**a**) and strains HM123 (WT), SG61 (Δ*trx1*), and IC76 (Δ*trx1* Δ*trx3*) (**b**) were transformed with plasmid p407.C169S and $H_2O_2$ treatments were performed and oxidation of the reporter estimated as described in Fig. 1. **c** Scheme depicting the role of cellular electron donors in roGFP2-Tpx1.C169S probe oxidation and reduction. Blockage of $H_2O_2$ detoxification either by lack of Tpx1 recycling or by lack of Ctt1 activity results in accumulation of basal or $H_2O_2$-induced oxidized roGFP2.C169S. Recycling of roGFP2.C169S depends on Grx1 at the expense of GSH and Pgr1. **d** Strains HM123 (WT) and EP160 (Δ*ctt1*) were transformed with plasmid p407.C169S and $H_2O_2$ treatments were performed and oxidation of the reporter estimated as described in Fig. 1. For each panel, data from three biological replicates are shown, with error bars (S.D.) displayed in Supplementary Fig. 6

the main $H_2O_2$ scavenger, Tpx1. We have represented probe oxidation upon ANT treatments with black lines, and we also used $H_2O_2$ treatments as controls (Fig. 5a, b and Supplementary Fig. 7). As shown in Fig. 5a, cytosolic HyPer is unable to sense mitochondrial ROS production upon ANT treatments.

We then treated wild-type cells expressing the roGFP2-Tpx1. C169S reporter with ANT. A small but significant increase of basal oxidation of the probe was observed 30–90 min after ANT exposure (Fig. 5b, top panel, black lines). It is worth mentioning that the growth rates of fission yeast cells grown in MM are not largely affected by ANT-dependent ETC inhibition, indicating that respiration is not the driving force of *S. pombe* proliferation

at high-glucose concentrations[26]. Oxidation of the roGFP2-Tpx1. C169S probe upon ANT treatment is much more noticeable in Δ*tpx1* cells, suggesting that $H_2O_2$ produced at the mitochondria and dispersed to the cytosol is rapidly scavenged by Tpx1 in wild-type cells (Fig. 5b, bottom panel).

Consistent with the chemical inhibition of the ETC with ANT, genetic blockage also triggers $H_2O_2$ leakage out of the membrane. Thus, strains carrying individual gene deletions of ETC components display growth defects on solid MM plates, which can be alleviated by the addition of antioxidants or oxygen depletion. We proposed that elevated steady-state levels of $H_2O_2$ due to electron leakage compromises the growth of these cells[27].

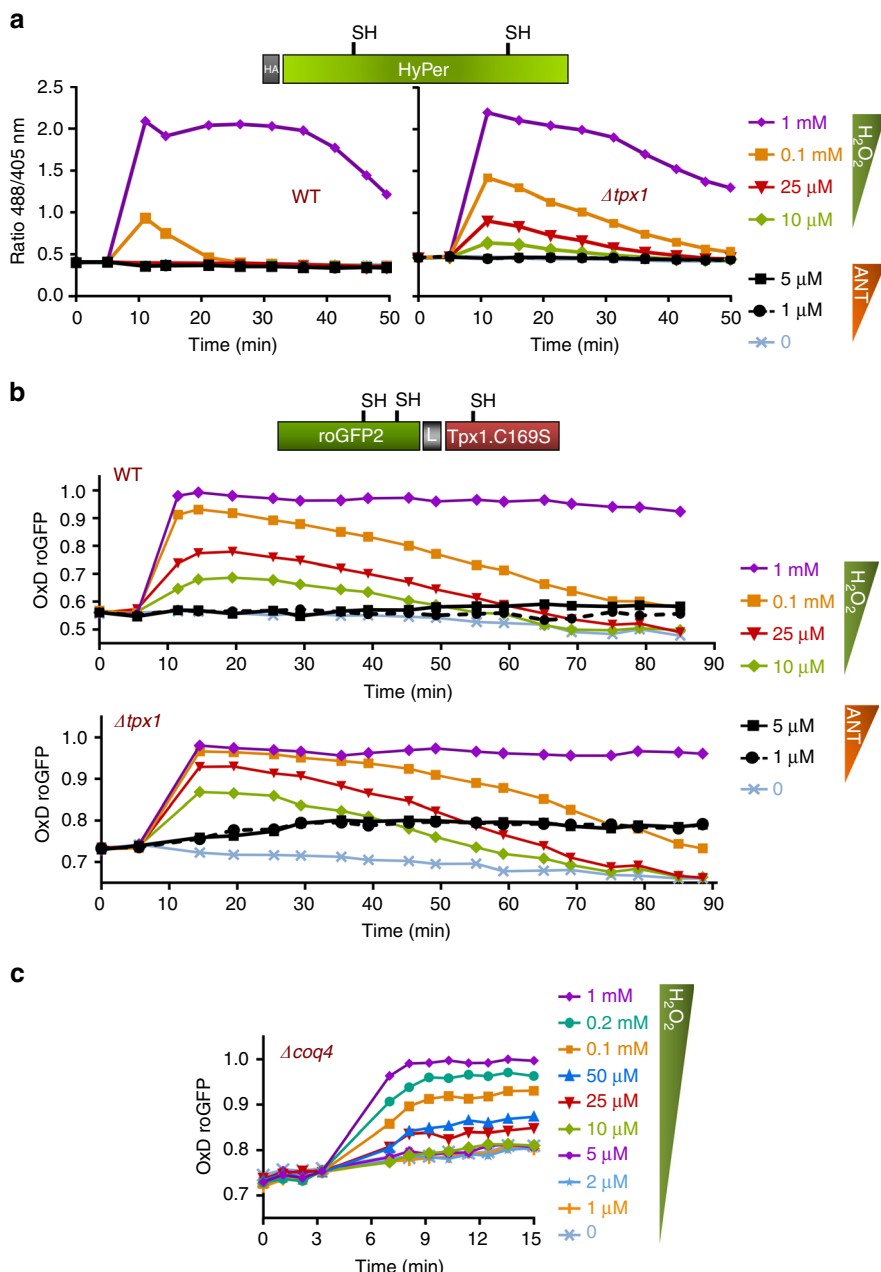

**Fig. 5** Cytosolic roGFP2-Tpx1.C169S can detect mitochondrial $H_2O_2$ production. Strains HM123 (WT) and SG5 (Δtpx1) were transformed with plasmids p605 (**a**) and p407.C169S (**b**). The indicated concentrations of $H_2O_2$ and antimycin A (ANT) were directly added to MM cultures and oxidation of the reporters was estimated as described in Fig. 1; oxidation of the probe upon ANT treatments is represented with solid (5 μM ANT) or dashed (1 μM ANT) black lines. **a** Data from two biological replicates are shown, with error bars (S.D.) displayed in Supplementary Fig. 7a. **b** Data from two (WT) or three (Δtpx1) biological replicates are shown, with error bars (S.D.) displayed in Supplementary Fig. 7b. **c** Strain Δcoq4 was transformed with plasmid p407.C169S and $H_2O_2$ treatments were performed and oxidation of the reporter estimated as described in Fig. 1. Data from three biological replicates are shown, with error bars (S.D.) displayed in Supplementary Fig. 7c

As shown now with the roGFP2-Tpx1.C169S probe, the steady-state levels of $H_2O_2$ in Δcoq4 cells, lacking a protein required for ubiquinone biosynthesis[27], drive 70–80% oxidation of the probe. This confirms the massive (and constant) $H_2O_2$ leakage from the mitochondria to the cytosol in this strain background, probably at the level of complex I[28] (Fig. 5c).

**Effect of glucose depletion on roGFP2-Tpx1.C169S oxidation.** The balance between respiration and fermentation largely depends on nutrient availability. In most cell types, including tumors and yeast, growth in nutrient-rich media induces fermentation and inhibits respiration (Crabtree effect[29,30]). Glucose-rich conditions inhibit respiratory metabolism in all cell types to different extents. Since ROS production is linked to mitochondrial metabolism, we decided to determine the effect of low glucose in roGFP2-Tpx1.C169S basal oxidation, $OxD_0$. Thus, we compared the $OxD_0$ in standard MM (MM-Glu), which contains 2% glucose as the only carbon source, with that in MM-Gly (1.85% glycerol and 0.15% glucose). To our surprise, $OxD_0$ is very low in MM-Gly (Fig. 6a and Supplementary Fig. 8), suggesting that the levels of intracellular $H_2O_2$ are lower in cells grown in respiratory-prone conditions.

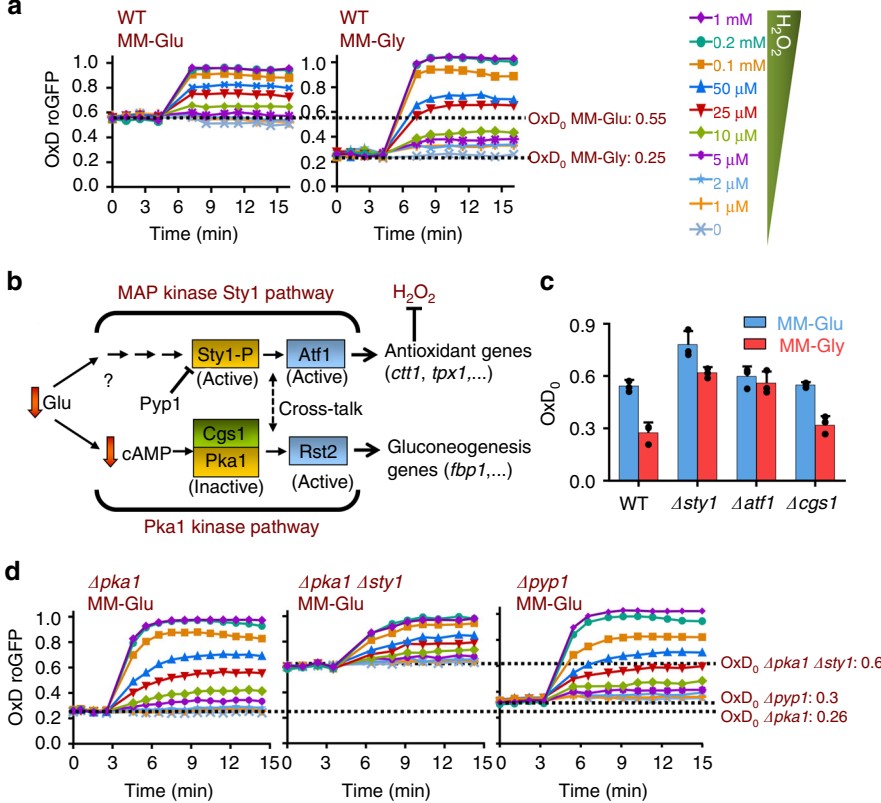

**Fig. 6** Effect of low-glucose media on roGFP2-Tpx1.C169S oxidation. **a** Strain HM123 transformed with p407.C169S was grown in MM-Glu or MM-Gly (respiratory-prone conditions), and $H_2O_2$ treatments were performed and oxidation of the reporter estimated as described in Fig. 1. Data from three biological replicates are shown, with error bars (S.D.) displayed in Supplementary Fig. 8a. **b** Scheme depicting the effect of glucose deprivation in the activation and inactivation of the Sty1 and Pka1 kinases, respectively. Activation of the Sty1 pathway triggers the transcription of Atf1-dependent antioxidant genes such as $ctt1$ or $tpx1$. Inactivation of the Pka1 kinase results in the de-repression of gluconeogenic genes and upregulation of mitochondrial oxygen consumption. The Pyp1 phosphatase is a negative regulator of the Sty1 kinase. **c** Strains HM123 (WT), EP16 ($\Delta sty1$), EP193 ($\Delta atf1$), and ED1150 ($\Delta cgs1$) were transformed with plasmid p407.C169S and basal $OxD_0$ levels determined as described in Fig. 1. Data from three biological replicates with error bars (S.D.) and individual data points overlaid (black dots) are shown. **d** Strains MC22 ($\Delta pka1$), MC24 ($\Delta pka1 \Delta sty1$), and AZ64 ($\Delta pyp1$) were transformed with plasmid p407.C169S and $H_2O_2$ treatments were performed and oxidation of the reporter estimated as described in Fig. 1. Data from three biological replicates are shown, with error bars (S.D.) displayed in Supplementary Fig. 8b

All cell types respond directly to glucose availability by activating signaling cascades. In *S. pombe*, the Pka1- and the Sty1-dependent gene expression programs are triggered upon glucose depletion, and are drivers of this glucose-dependent catabolite repression (Fig. 6b). The Pka1 kinase is specifically inactivated upon low-glucose conditions, and the downstream Rst2 transcription factor upregulates, among others, the gluconeogenic $fbp1$ gene and respiratory-related genes. Deletion of $pka1$ causes de-repression of $fbp1$[31], and high respiratory rates, as determined with oxygen consumption[32], even in the presence of elevated glucose levels. The Sty1 kinase and the Atf1 transcription factor are activated not only by glucose deprivation but also by many other environmental signals, and trigger a massive anti-stress gene program, which includes antioxidant activities[33].

Final steady-state ROS levels depend on the equilibrium between their synthesis and scavenging. We tested whether activation of the Sty1-Atf1 antioxidant cascade by glucose depletion could cause a decrease in basal $H_2O_2$ levels explaining the low $OxD_0$ of roGFP2-Tpx1.C169S in cells grown in respiratory-prone medium. We determined the $OxD_0$ of strains lacking components of the glucose-sensing Pka1 and Sty1 pathways, grown in standard MM (MM-Glu) or in MM-Gly. Unlike wild-type cells, strains lacking Atf1 display very similar levels of $OxD_0$ in both media (Fig. 6c), while cells lacking Cgs1,

an inhibitor of Pka1 kinase, behave as wild-type cells. Furthermore, cells lacking Pka1 display constitutive activation of the Sty1-Atf1 cascade through an unknown mechanism[32]. Concomitantly, the basal level of oxidation of the probe in MM-Glu is significantly lower in cells lacking Pka1 (Fig. 6d, left panel $OxD_0$: 0.26) than in wild-type cells (Fig. 6a, left panel; $OxD_0$: 0.55), in a Sty1-dependent manner (Fig. 6d, center panel, $OxD_0$ in $\Delta pka1 \Delta sty1$ cells: 0.6). These results suggest that upregulation of the Sty1-Atf1 antioxidant response causes the low levels of intracellular peroxides of strain $\Delta pka1$. Indeed, constitutive activation of only the MAP kinase Sty1 pathway, by deletion of its main phosphatase Pyp1, is sufficient to trigger low levels of oxidation of the roGFP2-Tpx1.C169S reporter in cells grown in MM-Glu (Fig. 6d, right panel, $OxD_0$ in $\Delta pyp1$ cells: 0.3).

**Monitoring mitochondrial activity with roGFP2-Tpx1.C169S.** As explained above, fission yeast grown in low-glucose concentrations shifts toward increased respiration, even though catabolite repression is not as dramatic as in other microbes[27,34]. It has been recently reported that glucose-dependent repression is more effective in standard YE medium than in MM, since oxygen consumption and ANT-dependent growth inhibition are lower in YE[27,35].

We searched for YE-based growth media altering mitochondrial activity to different extents. We determined oxygen consumption rates of wild-type cultures grown in standard YE rich media, containing 3% glucose, YE-0.08% glucose (this concentration of glucose in MM triggers a shift of fermentation-to-respiration in fission yeast, as nicely reported by Yanagida and colleagues[26]), and YE-3% glycerol (this media has been recently described to elicit an extreme carbon starvation response at the transcriptional level[36]). As shown in Fig. 7a, oxygen consumption at the same cell density is significantly lower in YE-3% glucose cultures, being maximal in YE-3% glycerol and intermediate in YE-0.08% glucose cell cultures. Furthermore, the presence of ANT largely compromises the growth of fission yeast in YE-0.08% glucose and YE-3% glycerol, but not in YE-3% glucose (Supplementary Fig. 9a).

We grew wild-type and $\Delta tpx1$ cells expressing roGFP2-Tpx1.C169S in the three media, and applied the ETC inhibitor ANT to determine whether enhanced mitochondrial activity could cause increased levels of $H_2O_2$ and concomitant probe oxidation. As shown before for MM-based media, the $OxD_0$ varied depending on the media (data not shown). To test the effect of ANT on probe oxidation in the three media, we monitored the percentage of oxidation from each starting $OxD_0$ levels. As shown in Fig. 7b–d and Supplementary Fig. 9bc, in both genetic backgrounds the effect of ANT treatment has a significantly larger effect on roGFP2-Tpx1.C169S oxidation when cells are grown in

YE-3% glycerol than in standard YE-3% glucose, with an intermediate oxidation in YE-0.08% glucose. We conclude that roGFP2-Tpx1.C169S oxidation after addition of ETC inhibitors can be used as readout of mitochondrial activity.

## Discussion

Redox biology is in need of reliable methods to measure in vivo fluctuations of $H_2O_2$. It may be quite easy to quantify large increases, which are normally linked to transient or permanent toxicity. Here, we demonstrate that the Prx-based sensor roGFP2-Tpx1.C169S can monitor differences in steady-state peroxide concentrations of strains carrying defects in the production or scavenging of $H_2O_2$. We recently described the gradients established across fission yeast membranes upon extracellular additions of peroxides[11]. Taking advantage of this knowledge, we can establish that our probe is able to detect intracellular $H_2O_2$ fluctuations in the low nanomolar (~10 nM) range, which is below the level where antioxidant redox signaling and toxicity occurs—at least 200–300 nM intracellular is required to trigger activation of the pre-toxic Pap1 signaling cascade[11].

We have demonstrated the specificity and sensitivity of roGFP2-Tpx1.C169S to detect subtle fluctuations of intracellular peroxides by modifying extracellular-to-intracellular peroxide gradients through overexpression of AQP8 in fission yeast. Two fluorescent $H_2O_2$ reporters, HyPer and a permeable dye, PY1-

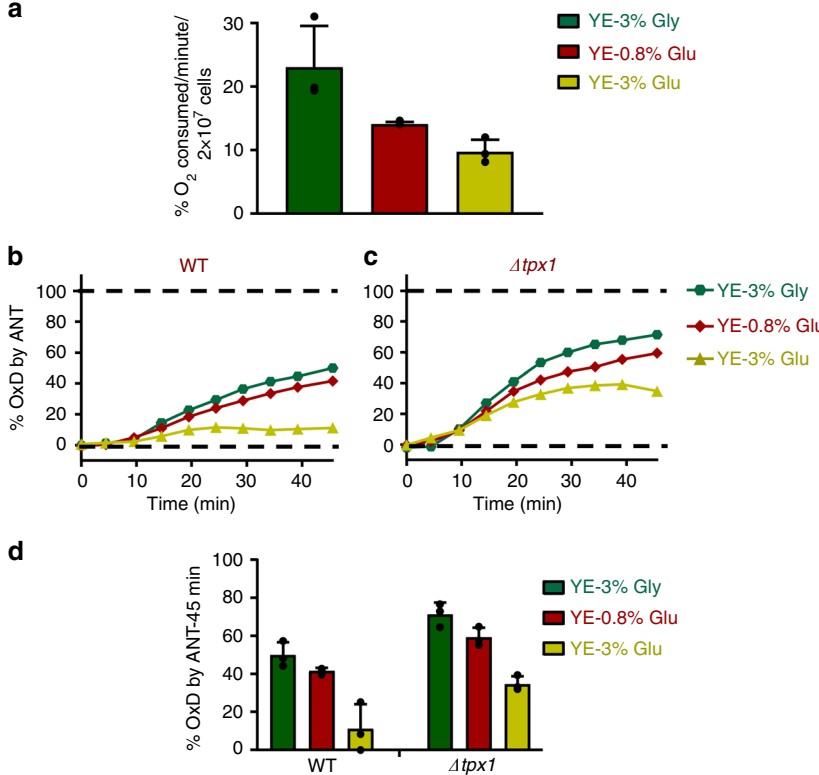

**Fig. 7** roGFP2-Tpx1.C169S oxidation by ANT is proportional to mitochondrial activity. **a** Oxygen consumption of *S. pombe* cells grown in different growth media. Wild-type strain 972 was grown on YE media with 3% glycerol (YE-3% Gly), 0.08% glucose (YE-0.08% Glu), or 3% glucose (YE-3% Glu), and when cultures reached an $OD_{600}$ of 0.5 oxygen consumption was recorded (see Methods). Data from three biological replicates with error bars (S.D.) and individual data points overlaid (black dots) are shown. **b**, **c** Strains HM123 (WT) (**b**) and SG5 ($\Delta tpx1$) (**c**) transformed with plasmid p407.C169S were grown in YE media containing 3% glycerol (YE-3% Gly—green circles), 0.08% glucose (YE-0.08% Glu—red diamonds), or 3% glucose (YE-3% Glu—yellow triangles). At an $OD_{600}$ of 1, cells were transferred to filtered MM, and probe oxidation upon ANT treatment (5 µM) was monitored for the times indicated. Ten minutes after addition of 1 mM $H_2O_2$ was used to determine 100% probe oxidation in each strain and condition, while 0% was the starting $OxD_0$ for each strain and condition. Data from three biological replicates are shown, with error bars (S.D.) displayed in Supplementary Fig. 9bc. **d** roGFP2-Tpx1.C169S oxidation upon ANT treatment at 45 min from data of panels (**b**) and (**c**) is displayed. Data from three biological replicates with error bars (S.D.) and individual data points overlaid (black dots) are shown

ME, had been used in mammalian cells to demonstrate that AQPs facilitate $H_2O_2$ entry after the addition of 20–50 μM $H_2O_2$ to the growing media[37,38]. Using our reporter roGFP2-Tpx1.C169S, we monitor cytosolic peroxide fluctuations upon 0.1–0.5 μM extracellular peroxides.

The extent of probe oxidation, $OxD_0$, can be used as a readout of steady-state levels of $H_2O_2$. Thus, roGFP2-Tpx1.C169S senses basal oxidative stress in cells lacking Tpx1, so that the percentage of basal roGFP2-Tpx1.C169S oxidation in Δtpx1 strain is 20% higher than that of wild-type cells (Fig. 3b). Simple calculations can correlate $OxD_0$ to intracellular steady-state levels of peroxides. In wild-type cells, 25–50 μM extracellular peroxides, which according to the 300:1 gradients corresponds to 0.08–0.16 μM intracellular $H_2O_2$, are able to oxidize the probe up to 75–85% (Fig. 3b, left panel). This degree of probe oxidation is observed under basal conditions in cells lacking Tpx1; we have recently reported that the toxic levels of peroxides observed in strain Δtpx1 are in the order of 0.3 μM[11].

By using this probe, we have demonstrated the role of moderate and/or severe $H_2O_2$ fluctuations in cell biology. For instance, we have confirmed that Tpx1 is the main peroxide scavenger at low concentrations of $H_2O_2$, while the role of catalase as a peroxide scavenger starts to be relevant at concentrations of applied peroxides ≥ 0.1 mM (Fig. 4d). We have also shown that reduction of roGFP2-Tpx1.C169S depends on the Grx/GSH system (Fig. 4a), and that probe reduction is sufficiently slow as to allow sensitive and sustained oxidation upon very mild peroxide concentrations. However, probe reduction timely reflects $H_2O_2$ scavenging in living cells: while upon 1 mM $H_2O_2$ extracellular stress wild-type cells can resume growth after ~5 h and roGFP2-Tpx1.C169S becomes reduced at the same time, cells lacking Ctt1 cannot resume growth when exposed to this level of oxidative stress and the probe remains oxidized for as long as 7 h (Fig. 4d).

Genetic alterations of the Trx or GSH/Grx systems can influence the redox state of roGFP2-Tpx1.C169S at the level of oxidation (in a Tpx1-dependen manner) or reduction, respectively (see Fig. 4c). This suggests that interventions known to affect not only $H_2O_2$ homeostasis but also the reduced-to-oxidized ratios of redoxins or GSH may affect roGFP2-Tpx1.C169S redox state.

During the course of our study, we have attempted to demonstrate mitochondrial ROS production during respiratory-prone conditions by depleting glucose from the media. As explained in the Results section, glucose-mediated catabolite repression is not dramatic in S. pombe, as it is in E. coli or S. cerevisiae. Thus, we have used MM-based (Fig. 6) and YE-based (Fig. 7) media with different concentrations of carbon sources to accomplish varying degrees of mitochondrial metabolism, and monitor their impact on roGFP2-Tpx1.C169S oxidation. We initially (Fig. 6) compared probe oxidation in cells grown in standard, high-glucose MM (MM-Glu) or in respiratory-prone MM-Gly. To our surprise, respiratory-prone media, known to induce mitochondrial activity, did not cause roGFP2-Tpx1.C169S oxidation but rather probe reduction (Fig. 6), suggesting that intracellular $H_2O_2$ levels are lower in cells grown in this media. Glucose depletion triggers, among others, the Sty1-Atf1 cascade and its antioxidant gene expression program[35]. In cells lacking the transcription factor Atf1, where the antioxidant program cannot be induced upon glucose deprivation, the $OxD_0$ in MM-Gly is not lower than in MM-Glu (Fig. 6c). On the contrary, cells lacking the Sty1 phosphatase Pyp1, or the kinase Pka1, where the antioxidant gene expression program is constitutively engaged even with high glucose, the $OxD_0$ of our probe in MM-Glu is very low, similar to that of a wild-type cells grown in MM-Gly (Fig. 6d). We conclude that glucose depletion causes the upregulation of the antioxidant gene expression program, and as a consequence, steady-state levels of peroxides are dampened, since scavenging prevails over mitochondrial $H_2O_2$ production.

An alternative strategy to promote mitochondrial $H_2O_2$ leakage different from lowering glucose from the media is the use of ETC inhibitors. Thus, we have demonstrated that addition of ANT, a complex III inhibitor which favors $H_2O_2$ release from the mitochondria, can cause cytosolic roGFP2-Tpx1.C169S oxidation. Furthermore, the extent of probe oxidation upon ANT addition can be used as readout of changes in mitochondrial activity due to glucose availability (Fig. 7). Thus, we have first measured oxygen consumption in three types of YE-based media to characterize their mitochondrial activities (Fig. 7a). The extent of probe oxidation upon inhibition by ANT is directly proportional to the mitochondrial activity in each media (Fig. 7b–d). However, this strategy does not allow the measurement of physiological mitochondrial $H_2O_2$ production unless the ETC inhibitor is added. Expression of the probe in the mitochondrial matrix or intermembrane space may be required to monitor $H_2O_2$ release upon massive ETC usage during mitochondrial respiration in the absence of ANT or other inhibitors, and experiments are underway.

In conclusion, we have exploited the use of fission yeast Tpx1.C169S as an exquisitely sensitive $H_2O_2$ sensor and signal transducer to roGFP. We have used the fission yeast model system to demonstrate the behavior of roGFP2-Tpx1.C169S, and provide numbers to redox biology events, such as mitochondrial $H_2O_2$ production, activation of signaling cascades and toxicity linked to enhanced $H_2O_2$ levels. We have also described that low-glucose-driven respiratory-prone conditions, known to stimulate mitochondrial activity, also trigger antioxidant signaling cascades, with a final decrease in cytosolic $H_2O_2$ levels. This indicates that steady-state levels of peroxides have to be experimentally determined to confirm or dismiss oxidative stress in a giving cellular context, since measuring only $H_2O_2$ production or scavenging may not be sufficient to anticipate which of both prevails. A remarkable example of complexity regarding glucose metabolism and $H_2O_2$ levels is tumor progression. Thus, cancer cells had been proposed to enhance not only glycolysis and glucose uptake, as proposed by Warburg in the 50s[39], but also mitochondrial metabolism, being respiration the main energy source in tumor cells (for excellent reviews on cancer metabolic reprogramming, see refs. [40–42]). Importantly, both mitochondrial $H_2O_2$ production and scavenging are enhanced in tumors compared with the surrounding normal cells[43]. Therefore, without in vivo measurements it is unpredictable to anticipate the steady-state levels of peroxides in cancer cells. Now, we expect our probe to be expressed and tested in other model systems, including human tumor cell lines, to find out whether the same intracellular $H_2O_2$ thresholds define the boundaries between aerobic metabolism, activation of antioxidant cascades, and toxicity linked to oxidative stress.

## Methods

**Fission yeast growth media and genetic manipulations.** For most of the experiments, cells were grown in filtered minimal medium (MM or MM-Glu, carrying 2% glucose) at 30 °C[44], supplemented with uracil, adenine, leucine, and/or cysteine in auxotrophic strains. We also used, when indicated, MM-Gly, containing as carbon source 1.85% glycerol and 0.15% glucose. Yeast extract (YE) rich medium or autoclaved MM could not be used for direct fluorescence measurements in the plate reader, since they display strong interference with intracellular fluorescence. When indicated, cells were grown in rich medium (YE) with various concentrations and sources of carbon supplies: 3% glucose, 0.08% glucose, or 3% glycerol.

**Generation of plasmids and strains.** All the plasmids described here express fluorescent reporters under the control of the constitutive *sty1* promoter[22], and most of them are episomal, with an average of 7–8 plasmid copies per cell[45]. According to previous quantifications of other proteins expressed from the same promoter[11], the intracellular protein concentration arising from these constructs is

in the order of 2–10 μM. HyPer and Grx1-roGFP2 were PCR-amplified from the original pHyPer-Cyto (Evrogen) and pLPCX cyto Grx1-roGFP2[21] plasmids, and cloned after a HA-coding sequence, yielding plasmids p605 and p379; these plasmids allowed the expression of HA-HyPer and HA-Grx1-linker (L)-roGFP2, respectively. We also PCR-amplified from pLPCX cyto Grx1-roGFP2 the DNA sequences coding for roGFP2 open reading frame and for the intermediate linker (L), coding for a 30 amino acids-long glycine-rich domain, and cloned them in tandem yielding plasmid p406, coding for roGFP2-L. This plasmid was used as a backbone to clone three PCR-amplified open reading frames from *S. pombe*, yielding plasmids p429, p407, and p407.C169S, allowing the expression of roGFP2-L-Gpx1, roGFP2-L-Tpx1, and roGFP2-L-Tpx1.C169S, respectively. p406 was also used to clone a synthetic open reading frame (supplied by IDT Technologies) coding for Tsa2ΔC$_R$, yielding plasmid p676 coding for roGFP2-Tsa2ΔC$_R$. We also generated an integrative plasmid for the expression of roGFP-Tpx1.C169S under the control of the constitutive *sty1* promoter, named p504′.C169S. To express human AQPs in fission yeast, we ordered synthetic genes coding for AQP8 and AQP4 flanked by an endoplasmic reticulum driving coding sequence at 5′, and Myc and FLAG tag coding sequence at 3′ (IDT Technologies), and cloned them in the *ura4* episomal vector pREP.42x plasmid[46], with the *nmt* promoter substituted with the *sty1* promoter, yielding p675 (AQP8) and p682 (AQP4). The absence of mutations in each of these plasmids was confirmed by DNA sequencing. The genotypes of the strains used in this study are shown in Supplementary Table 1. Some of these strains were constructed for this study as follows. Strain SG63 was derived from the Δ*pap1* strain of a *S. pombe* haploid deletion collection (Bioneer), by replacing the *kanMX6* with *natMX6* cassette. Strain AD88 was constructed by crossing AD84[11] (h⁻ *pgr1::natMX6*) with JA212 (h⁺ *leu1-32;* our laboratory stock) and selecting clones in MM plates supplemented with leucine. Strain SG60 was obtained by replacing *kanMX6* cassette from strain MJ15[47] (h⁺ *trx1::kanMX6*) with *natMX6* cassette. Strains SG61 and EP16 were obtained by insertion of an *ura4* cassette in the *trx1* and *sty1* loci, respectively, of strain PN513 (h⁻ *ura4-D18 leu1-32;* our laboratory stock). Strain SG260 was obtained by mating MJ15[47] (h⁺ *trx1::kanMX6*) with SG257[48] (h⁻ *trx3::natMX6*). Strain EP160 was obtained by crossing CN513[49] (h⁻ *ctt1::ura4 ade6-M216 leu1-32 ura4-D18*) with JA368 (h⁺ *ura4-D18 leu1-32;* our laboratory stock) and selecting clones in MM supplemented with leucine but lacking uracile. Strain AZ81 was obtained by crossing AZ61 (h⁺ *pka1::kanMX6 sty1::ura4*, obtained by insertion of an *ura4* cassette in the *sty1* locus of a *pka1::kanMX6* strain from Bioneer collection) with JA365 (h⁻ *ura4-D18;* our laboratory stock) and selecting clones in MM plates. The Δ*pyp1* strain of the Bioneer *S. pombe* haploid deletion collection was crossed with 972 to yield strain AZ64 after selection in plates supplemented with leucine. Strains HM123 (wild-type), SG5 (Δ*tpx1*), Δ*coq4*, AD88 (Δ*pgr1*), SG61 (Δ*trx1*), IC76 (Δ*trx1 Δtrx3*), EP160 (Δ*ctt1*), MC22 (Δ*pka1*), and MC24 (Δ*pka1 Δsty1*) were transformed with the episomal plasmids described above following standard genetic techniques[50], and transformants were selected in MM plates lacking leucine; only strain SB36 (Δ*grx1*), without leucine auxotrophy, was crossed with HM123 carrying p407. C169S, and plasmid acquisition was selected by fluorescence. The integrative plasmid p504′.C169S was inserted at the *leu1-32* loci of strain SG5 (Δ*tpx1*), yielding strain MC167.C169S.

**Growth of strains expressing HyPer or roGFP derivatives.** For wild-type backgrounds, standard MM-based early stationary phase pre-cultures were diluted in filtered MM (either MM-Glu or MM-Gly), or in YE-based media (see above) to reach an OD$_{600}$ of 1 after 4–5 duplications. In the case of some strain backgrounds displaying defects to grow under aerobic conditions, such as Δ*tpx1, Δpgr1*, and Δ*coq4*, pre-cultures were grown under anaerobic conditions, using capped-bottles filled to the top with MM medium at 30 °C without shaking; these anaerobic pre-cultures were then diluted in normal aerobic MM-containing flasks (Δ*tpx1*) or anaerobic capped-bottles (Δ*pgr1* and Δ*coq4*), till cultures reached an OD$_{600}$ of 1 after 3–4 duplications. For strains Δ*trx1* and Δ*trx1 Δtrx3*, auxotrophic for cysteine, MM aerobic cultures contained 38 μg/ml cysteine. The fluorescence of 190 μl of these cultures, at an OD$_{600}$ of 1, was directly monitored in 96-well plates in a Fluorstar OMEGA (BMG Labtech) as described below. In the case of strain MC98. C169S, expressing low levels of roGFP2-Tpx1.C169S from an integrative plasmid, MM cultures at an OD$_{600}$ of 1 obtained as above where gently centrifuged at 1000 × *g* for 3 min at room temperature, and 75% of the supernatant was withdrawn; the remaining buffer was used to resuspend the cell pellets reaching an OD$_{600}$ of 4; this concentrated cultures were monitored in the fluorescence plate reader. In the case of strains growing in YE-based media, cell cultures were centrifuged and resuspended in the same volume of standard MM to allow fluorescence readings.

**In vivo measurement of HyPer and roGFP-derivatives oxidation.** HyPer has two excitation maxima at 420 and 500 nm and one emission peak at 516 nm[14]. Similarly, roGFP2 exhibits two excitation maxima at 400 nm and 475–490 nm when fluorescence emission is monitored at 510 nm. Either HyPer or roGFP2 oxidation was monitored using a fluorescence plate reader FLUOstar OMEGA (BMG Labtech). In both cases, we used excitation filters of 400-10 and 485BP12, combined with emission filter EM520. For each strain to be analyzed, we transferred 190 μl of the MM culture at an OD$_{600}$ of 1 (strains carrying episomal plasmids) or 4 (strains with integrative plasmids) to as many wells of a 96-well

imaging plate (Krystal Microplate™ 215003, Porvair Sciences) as experimental treatments were to be tested. Recording at the two excitation wavelengths was initiated, with four readings every 1–1.5 min, with continuous shaking at 30 °C. After these four cycles, 10 μl of MM containing or not the different treatments were added to the wells to accomplish the final concentrations of reagents indicated in the figures, and recording rapidly proceeded. To one well, the oxidant N,N,N′,N ′-tetramethylazodicarboxamide (diamide) was added to a final concentration of 5 mM, and to another well the reductant dithiothreitol (DTT) was added to a final concentration of 50 mM. These samples served as fully oxidized and fully reduced sensor controls for roGFP derivatives (see below); for some strains (Δ*pgr1, Δcoq4*, and Δ*trx1 Δtrx3* strains) or growth conditions (MM-Gly and low glucose YE cultures), treatment with 1 mM H$_2$O$_2$ were used as fully oxidized sensor references instead of diamide. *S. pombe* cells displayed basal levels of fluorescence at 510 nm after excitation at either 400 or 485 nm, and this basal fluorescence changed with diamide and DTT treatments; therefore, for each strain expressing HyPer or roGFP derivatives, we grew cultures of the same genetic background lacking the plasmids, and performed the same treatments on the 96-well plates as to their genetic counterparts expressing the chimeras; then, after recording, for each specific treatment we subtracted the fluorescence values of the empty strain to those of the strain expressing the reporter. In the case of YE-based cultures, in which we had to change the culture media for fluorescence recording, we did not subtract the fluorescence values of the empty strains. For HyPer, diamide and DTT only had modest effects in oxidation and reduction of the probe, and could not be used as controls of maximal oxidation/reduction; therefore, we only represent the data as ratio 488/405 nm. For roGFP derivatives, we determined the degree of sensor oxidation (OxD) with the following Eq. (1) as described[51]:

$$\text{OxD (per 1)} = \frac{(I_{\text{sample 488}} \times I_{\text{DTT 405}}) - (I_{\text{sample 405}} \times I_{\text{DTT 488}})}{(I_{\text{sample 405}} \times I_{\text{diamide 488}}) - (I_{\text{sample 405}} \times I_{\text{DTT 488}}) - (I_{\text{sample 488}} \times I_{\text{diamide 405}}) + (I_{\text{sample 488}} \times I_{\text{DTT 405}})}$$

(1)

where *I* represents the fluorescence intensity at 510 nm after excitation at either 405 nm or 488 nm of the sample treated with H$_2$O$_2$ at a given time and concentration, the control values intensity after diamide or DTT treatments were the fluorescence intensity values after 5–10 min of these control treatments. When indicated (Fig. 7, YE-based cultures) the OxD values of the cultures were expressed in a 0-to-100% scale, 0 being the starting OxD for each strain in the specific culture media and 100% being the maximum level of oxidation upon 1 mM H$_2$O$_2$ 10 min after stress imposition. All experiments were performed in biological triplicates (or duplicates, when indicated) using cells obtained from three independent cell cultures. For each new strain to be tested, HM123 carrying plasmid p407.C169S was always added and analyzed in the same 96-well plate, to use it as an internal control of basal probe oxidation.

**Oxygen sensitivity assay on solid plates.** For survival on solid plates, *S. pombe* strains were grown, diluted and spotted on MM plates and plates were incubated at 30 °C for 3–4 days. To grow cells in solid media in an anaerobic environment, we incubated the plates at 30 °C in a tightly sealed plastic bag containing a water-activated Anaerocult A sachet (Merck, Darmstadt, Germany)[20], or in a nitrogen-filled anaerobic chamber (Forma Anaerobic System, Thermo Electron Corp.).

**Measurement of oxygen consumption.** Oxygen consumption of 2 × 10⁷ cells in 1 ml was performed as described before[27]. Briefly, cells were grown in YE media containing different concentrations of carbon sources till they reached an OD$_{600}$ of 0.5. Cells were harvested, and resuspended in 1 ml of standard MM to a final OD600 of 1. The measurements were made using a Hansatech Oxygraph (Hansatech), with readings being recorded during 10 min. Each one of the measurements was performed from biological triplicates.

**Growth curves.** Cells from cultures of wild-type strain 972 in standard YE at an OD$_{600}$ of 1 were harvested and resuspended in YE with different carbon sources to an OD$_{600}$ of 0.1. Cell growth was monitored using an assay based on automatic measurements of optical densities, as previously described[52]. Briefly, for each resuspended culture we placed 100-μl samples, with or without ANT (5 μM), into 96-well non-coated polystyrene microplates with an adhesive plate seal. Each experimental condition was measured in duplicate (technical replicates). We used Power Wave microplate scanning spectrophotometer (Bio-Tek) to obtain the growth curves. Incubation temperature was kept at 30 °C, the microplates were subjected to continuous shaking and the readings were done every 10 min during a 48 h period. The OD$_{600}$ was automatically recorded using Gen5 software. Each one of the growth curves was performed from biological triplicates.

**Statistics.** For in vivo fluorescence probe oxidation quantification, OxD, one independent culture of the strain of interest was grown for each replicate. In all figure panels, values of mean of $n = 3$ are indicated, with the mean ± s.d displayed in supplementary figures. Only in some specific figure panels, results from biological duplicates are shown. The exact $n$ value is described in each figure legend. In the case of spot assays, 3-to-4 independent experiments with very similar results were performed.

**Reporting summary.** Further information on research design is available in the Nature Research Reporting Summary linked to this article.

## Data availability
The data that support the findings of this study are available from the corresponding author upon reasonable request.

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

## Acknowledgements
We acknowledge Rosa Bretón and Santiago Lamas for providing plasmid pLPCX Grx1-roGFP2, and Guy Lenaers for strain Δcgs1. This work is supported by the Ministerio de Ciencia, Innovación y Universidades (Spain), PLAN E and FEDER (BFU2015-68350-P and PGC2018-093920-B-I00 to E.H.) and by Unidad de Excelencia María de Maeztu (MDM-2014-0370). The Oxidative Stress and Cell Cycle group is also supported by Generalitat de Catalunya (Spain) (2017-SGR-539). E.H. is recipient of an ICREA Academia Award (Generalitat de Catalunya, Spain).

## Author contributions
M.C., L.C., E.B. and M.M.-B. performed most experiments. M.C., L.C., S.B., I.M.-F., R.S., J.A. and E.H. analyzed the data. E.H. wrote the paper.

## Competing interests
The authors declare no competing interests.
