## [Peer Review File · Nature Communications]

Editorial Note: This manuscript has been previously reviewed at another journal that is not operating a transparent peer review scheme. This document only contains reviewer comments and rebuttal letters for versions considered at *Nature Communications*. Mentions of the other journal and of prior referee reports have been redacted.

REVIEWERS' COMMENTS:

Reviewer #1 (Remarks to the Author):

To achieve the goal of general interest to others in the community, the focus in this manuscript needs to be further shifted from making and characterizing a new probe that is similar in design and performance with an existing one. Pages 1-11 aren't of general interest given the existing literature.

From the authors' response:

[REDACTED]

The interventions that have been demonstrated are drastic:

1. genetic knockouts of major antioxidants and introduction of peroxiporin channels within the membranes changed extracellular/intracellular gradients as could be anticipated (Antunes & Cadenas, 2000, FEBS 475, 121-126)

and

2. chemical inhibitors of the electron transport chain led to changes in oxidation of the probe in the cytosol, particularly when cytosolic antioxidants were knocked out.

The experiments with different carbon sources are new, and the further explanations of ongoing work that the authors provided in the rebuttal letter are interesting. These findings constitute a small fraction of the manuscript, however (the last figures), and the findings aren't explained and considered in as much depth in the text of the manuscript as in the rebuttal letter. The latter part of the study could perhaps be expanded and presented for a more general audience, with attention to which findings are specific to pombe and which may find more general applicability.

As written, whether the authors intend it or not, the new and interesting biological results don't get the space, clarity, or discussion that they deserve.

Reviewer #2 (Remarks to the Author):

In this manuscript Carmona and co-workers report on the development of a genetically encoded H₂O₂ sensor, roGFP2-Tpx1.C169S. This probe is based on a fusion of a typical 2-Cys peroxiredoxin, Tpx1, from the fission yeast *S. pombe* to a redox-sensitive green fluorescent protein. Carmona et al, then characterised the function of this construct in *S. pombe* and have used it to gain novel insights into H₂O₂ dynamics in living *S. pombe* cells, including changes in cytosolic H₂O₂ dependent upon changes in carbon source, respiratory chain activity and metabolic signalling.

I previously reviewed this manuscript for **[REDACTED]** and consistent with my review of the manuscript on that occasion, it remains my opinion that all experiments appear to have been

performed to a high standard, with appropriate controls. Characterization of the probe is appropriate, and the results are consistent with known data in the literature. The authors' interpretations and analyses seem to be correct.

My major comment to the previous version of this manuscript was the level of novelty. A concern which was apparently shared by the other two reviewers. It was my opinion that there should be a greater emphasis placed on the application of the probe to investigate new biology rather than simply on reporting and characterising another sensor, which in its mechanistic principle is very similar to the previously reported roGFP2-Tsa2d_Cr sensor. In particular, I found the first results in the direction of H₂O₂ changes dependent upon respiratory chain activity and metabolic signalling to be intriguing. I think it is fair to say that H₂O₂ handling and its coupling to metabolism is still poorly understood, even in simple eukaryotes, such as *S. pombe*. In the new version of the manuscript the authors have now added some extra data in this regard. Their data show that H₂O₂ levels are not simply determined by respiratory chain activity and the quantity of electrons leaking from the respiratory chain. Rather, metabolic signalling and the consequent change in the expression levels of many different redox (and other) proteins is also extremely important. The authors clearly demonstrate that this can lead to situations where cells with very low respiratory chain activity have much higher H₂O₂ levels than cells with more active respiratory chains. In my opinion, this may well (probably will) have implications far beyond the *S. pombe* system. However, I would be tempted to sell/promote this point even more (see also my comment below).

My recommendation to the editor regarding the previous version of this manuscript at **[REDACTED]** was to accept for publication provided that the authors could provide extra insights from the application of their probe, particularly in the direction of metabolic signalling and to shift the focus away from simple probe characterisation. Although in the revised manuscript I feel this could still be taken further, overall, I recommend publication of this manuscript in Nature Communications.

Major comment:

The discussion is somewhat repetitive of the results section. The authors could use discussion as an opportunity to discuss some of their new insights in a broader context. This would surely also help to shift the focus of the study from one of probe characterisation to one of probe application and new biological insights. Their response to some of my previous comments regarding the metabolic signalling aspects was particularly helpful: could they summarise some of this information in the discussion, setting it in the context of their findings? It could also be interesting for the broader audience to discuss potential implications of their findings for other cells types that are thought to be prone to aerobic glycolysis, for example some tumour cells.

Minor Comments:

End of page 6, beginning of page 7: As the authors describe it anyway, it would be good to more clearly describe the benefit of the resolving cysteine mutation here. Its major effect is to reduce or eliminate thioredoxin-mediated reduction of the peroxiredoxin. However, and this is the most important point, it does not seem to affect roGFP2-mediated reduction, thus resolving cysteine mutation eliminates thioredoxin competition with regards to peroxiredoxin reduction and thereby dramatically enhances the efficiency of oxidation transfer to roGFP2.

Bruce Morgan

RESPONSE TO REVIEWERS' COMMENTS

NCOMMS-19-2808837-T

Carmona et al.

Monitoring cytosolic H₂O₂ fluctuations arising from altered plasma membrane gradients or from mitochondrial activity

We are very thankful to the reviewers for their fair and careful consideration of our manuscript, for the time they invested in the revision process, and for their kind words about our work. We have introduced all the changes they have suggested.

Reviewer #1

To achieve the goal of general interest to others in the community, the focus in this manuscript needs to be further shifted from making and characterizing a new probe that is similar in design and performance with an existing one. Pages 1-11 aren't of general interest given the existing literature.

From the authors' response:

[REDACTED]

The interventions that have been demonstrated are drastic:

1. genetic knockouts of major antioxidants and introduction of peroxiporin channels within the membranes changed extracellular/intracellular gradients as could be anticipated (Antunes & Cadenas, 2000, FEBS 475, 121-126)

and

2. chemical inhibitors of the electron transport chain led to changes in oxidation of the probe in the cytosol, particularly when cytosolic antioxidants were knocked out.

The experiments with different carbon sources are new, and the further explanations of ongoing work that the authors provided in the rebuttal letter are interesting. These findings constitute a small fraction of the manuscript, however (the last figures), and the findings aren't explained and considered in as much depth in the text of the manuscript as in the rebuttal letter. The latter part of the study could perhaps be expanded and presented for a more general audience, with attention to which findings are specific to pombe and which may find more general applicability.

As written, whether the authors intend it or not, the new and interesting biological results don't get the space, clarity, or discussion that they deserve.

We agree with the reviewer that we spent much more time and space in the rebuttal letter than in the Discussion of the manuscript to describe the effect of ETC inhibitors and glucose depletion media on probe oxidation. We have now increased the explanations in the Discussion section regarding ANT treatment, OxD₀ values and effect of glucose media on probe oxidation, so that most of the description of our previous response to reviewers is provided to the general audience. We have also reduced some contents of the first part of the manuscript, and eliminated Supp. Fig 4a. However, I have not eliminated any major section, since that may be of relevance to the other reviewer and to the general readers, as it is to us.

Reviewer #2

In this manuscript Carmona and co-workers report on the development of a genetically encoded H₂O₂ sensor, roGFP2-Tpx1.C169S. This probe is based on a fusion of a typical 2-Cys peroxiredoxin, Tpx1, from the fission yeast *S. pombe* to a redox-sensitive green fluorescent protein. Carmona et al, then characterised the function of this construct in *S. pombe* and have used it to gain novel insights into H₂O₂ dynamics in living *S. pombe* cells, including changes in cytosolic H₂O₂ dependent upon changes in carbon source, respiratory chain activity and metabolic signalling.

I previously reviewed this manuscript for [REDACTED] and consistent with my review of the manuscript on that occasion, it remains my opinion that all experiments appear to have been performed to a high standard, with appropriate controls. Characterization of the probe is appropriate, and the results are consistent with known data in the literature. The authors' interpretations and analyses seem to be correct.

My major comment to the previous version of this manuscript was the level of novelty. A concern which was apparently shared by the other two reviewers. It was my opinion that there should be a greater emphasis placed on the application of the probe to investigate new biology rather than simply on reporting and characterising another sensor, which in its mechanistic principle is very similar to the previously reported roGFP2-Tsa2d_Cr sensor. In particular, I found the first results in the direction of H₂O₂ changes dependent upon respiratory chain activity and metabolic signalling to be intriguing. I think it is fair to say that H₂O₂ handling and its coupling to metabolism is still poorly understood, even in simple eukaryotes, such as *S. pombe*. In the new version of the manuscript the authors have now added some extra data in this regard. Their data show that H₂O₂ levels are not simply determined by respiratory chain activity and the quantity of electrons leaking from the respiratory chain. Rather, metabolic signalling and the consequent change in the expression levels of many different redox (and other) proteins is also extremely important. The authors clearly demonstrate that this can lead to situations where cells with very low respiratory chain activity have much higher H₂O₂ levels than cells with more active respiratory chains. In my opinion, this may well (probably will) have implications far beyond the *S. pombe* system. However, I would be tempted to sell/promote this point even more (see also my comment below).

My recommendation to the editor regarding the previous version of this manuscript at [REDACTED] was to accept for publication provided that the authors could provide extra insights from the application of their probe, particularly in the direction of metabolic signalling and to shift the focus away from simple probe characterisation. Although in the revised manuscript I feel this could still be taken further, overall, I recommend publication of this manuscript in Nature Communications.

Major comment:

The discussion is somewhat repetitive of the results section. The authors could use discussion as an opportunity to discuss some of their new insights in a broader context. This would surely also help to shift the focus of the study from one of probe characterisation to one of probe application and new biological insights. Their response to some of my previous comments regarding the metabolic signalling aspects was particularly helpful: could they summarise some of this information in the discussion, setting it in the context of their findings? It could also be interesting for the broader audience to discuss potential implications of their findings for other cells types that are thought to be prone to aerobic glycolysis, for example some tumour cells.

Thanks a lot to the reviewer for his in-depth revision of our work, and for his fair and kind comments. We did not intend to write a repetitive Discussion, and we tend to write short ones to allow free interpretation by the readers. But we have now, upon request by the reviewers, expanded the explanations in the Discussion section regarding ANT treatment, OxD₀ values and

effect of glucose media on probe oxidation. Now, many of the details which I provided in the previous response-to-reviewers letter are included in the Discussion.

Regarding tumor progression, the reviewer is sure aware of the amazing complexity of metabolic reprogramming and redox balance changes occurring in cancer cells. My naïf view based on existing literature, goes as follows:

- 1) Regarding metabolic reprogramming: Otto Warburg proposed in the 50s (Science 1956, 123:309; Science 1956, 124:269) that aerobic glycolysis (and glucose uptake) was highly up-regulated in cancer cells. However, it is now clear that the main role of glycolysis in cancer progression is not ATP generation, but rather the supply of precursors for anabolic pathways. Indeed, mitochondrial oxidative phosphorylation (OXPHOS) and ATP production through the ETC is also exacerbated in tumor cells, being fuelled from glycolytic precursors, from fatty acids and from glutamine. Thus, it has been proposed that cancer cells do not switch from respiration to fermentation, but rather activate them both (for a recent review by the Chandel lab, see Sci. Adv. 2016; 2 : e1600200).
- 2) Regarding ROS synthesis, it is also widely accepted now that localized ROS production from the mitochondria or from NOXs is enhanced in cancer cells compared to normal cells, probably due to the exacerbated OXPHOS metabolism and to specific oncoproteins. This triggers tumorigenic pathways which facilitate tumor progression.
- 3) Regarding ROS scavenging: to avoid excessive ROS levels which could be toxic, cancer cells also have higher levels of ROS scavenging activities, mainly due to constitutive activation of the transcription factor NRF2.

Furthermore, ROS levels in cancer cells may depend on:

- 1) cancer type
- 2) cell type (heterogeneity of cells in a given tumor)
- 3) cancer timing (starting, proliferating, metastasis stages)
- 4) sub-cellular localization (i.e. close to a NOX)

I would not feel comfortable writing extensively about ROS and tumors, since it is far from my area of expertise. However, I have modified the last paragraph of the Discussion as follows:

“In conclusion, we have exploited the use of fission yeast Tpx1.C169S as an exquisitely sensitive H₂O₂ sensor and signal transducer to roGFP. We have used the fission yeast model system to demonstrate the behavior of roGFP2-Tpx1.C169S, and provide numbers to redox biology events, such as mitochondrial H₂O₂ production, activation of signaling cascades and toxicity linked to enhanced H₂O₂ levels. We have also described that low glucose-driven respiratory-prone conditions, known to stimulate mitochondrial activity, also trigger antioxidant signaling cascades, with a final decrease in cytosolic H₂O₂ levels. This indicates that steady-state levels of peroxides have to be experimentally determined to confirm or dismiss oxidative stress in a given cellular context, since measuring only H₂O₂ production or scavenging may not be sufficient to anticipate which of both prevails. A remarkable example of complexity regarding glucose metabolism and H₂O₂ levels is tumor progression. Thus, cancer cells had been proposed to enhance not only glycolysis and glucose uptake, as proposed by Warburg in the 50s³⁹, but also mitochondrial metabolism, being respiration the main energy source in tumor cells (for excellent reviews on cancer metabolic reprogramming, see⁴⁰⁻⁴²). Importantly, both mitochondrial H₂O₂ production and scavenging are enhanced in tumors compared to the surrounding normal cells⁴³. Therefore, without *in vivo* measurements it is unpredictable to anticipate the steady-state levels of peroxides in cancer cells. Now, we expect our probe to be expressed and tested in other model systems, including human tumor cell lines, to find out whether the same intracellular H₂O₂ thresholds define the boundaries between aerobic metabolism, activation of antioxidant cascades, and toxicity linked to oxidative stress”.

Minor Comments:

End of page 6, beginning of page 7: As the authors describe it anyway, it would be good to more clearly describe the benefit of the resolving cysteine mutation here. Its major effect is to reduce or eliminate thioredoxin-mediated reduction of the peroxiredoxin. However, and this is the most important point, it does not seem to affect roGFP2-mediated reduction, thus resolving cysteine mutation eliminates thioredoxin competition with regards to peroxiredoxin reduction and thereby dramatically enhances the efficiency of oxidation transfer to roGFP2.

Thanks for the suggestion. We have now added the following sentence in the description of the generation of this probe:
"Therefore, this mutation eliminates Trx competition and may enhance the oxidation transfer from Cys48-SOH to roGFP2".